

# A path planning method using modified harris hawks optimization algorithm for mobile robots

Cuicui Cai, Chaochuan Jia, Yao Nie, Jinhong Zhang and Ling Li

College of Electronics and Information Engineering, West Anhui University, Lu'an, China

## ABSTRACT

Path planning is a critical technology that could help mobile robots accomplish their tasks quickly. However, some path planning algorithms tend to fall into local optimum in complex environments. A path planning method using a modified Harris hawks optimization (MHHO) algorithm is proposed to address the problem and improve the path quality. The proposed method improves the performance of the algorithm through multiple strategies. A linear path strategy is employed in path planning, which could straighten the corner segments of the path, making the obtained path smooth and the path distance short. Then, to avoid getting into the local optimum, a local search update strategy is applied to the HHO algorithm. In addition, a nonlinear control strategy is also used to improve the convergence accuracy and convergence speed. The performance of the MHHO method was evaluated through multiple experiments in different environments. Experimental results show that the proposed algorithm is more efficient in path length and speed of convergence than the ant colony optimization (ACO) algorithm, improved sparrow search algorithm (ISSA), and HHO algorithms.

## INTRODUCTION

A mobile robot is a complex autonomous system with multiple sensors that implement hazardous and repetitive tasks in a specific environment, and it has been widely used in areas, such as industry, agriculture, medicine, and the military (*Orozco-Rosas, Montiel & Sepúlveda, 2019*; *Pattnaik, Mishra & Panda, 2021*). In recent years, many researchers have been investigating mobile robots in several directions, including system control, path planning, simultaneous localization navigation, and trajectory tracking. Path planning is an important component of mobile robots and plays a key role in accomplishing tasks. As a whole, path planning intends to design a path that avoids obstacles from the initial position to the target position under specific constraints (*Deng et al., 2021*; *Yang et al., 2022*).

Many conventional path-planning methods have been presented such as D-star algorithm (D*), A-star algorithm (A*), and probabilistic road map (PRM) algorithm (*Patle et al., 2019*). *Maurovic et al. (2018)* discussed the path design problem and presented a D-star algorithm for path planning in a mobile robot, which had high performance in a dynamic environment and dynamically changing localization requirements. *Liu et al.*

Corresponding author
Cuicui Cai, caicuihappy@wxc.edu.cn

*(2021)* developed an improved A-star algorithm for mobile robot path planning, and the algorithm achieved high-quality paths in rectangular obstacle environments. *Wang & Cai (2018)* used a PRM algorithm for nuclear facilities' path planning, and the proposed method has been proven to be effective in path design within a radioactive environment. Traditional algorithms have several advantages, such as simple principles and easy implementation, and have been employed for mobile robots. However, in a complex situation, these methods have a slow convergence rate and the generated paths may be too rough to satisfy the optimal path (*Orozco-Rosas et al., 2022*; *Patle et al., 2019*).

To address the shortcomings of the above-mentioned traditional methods, many metaheuristic algorithms have been presented recently. A method of path planning using an improved ant colony optimization (ACO) algorithm is presented by *Akka & Khaber (2018)*, and the results indicated that the obtained optimum path outperformed the conventional algorithms. *Song, Pan & Chu (2020)* developed an improved cuckoo search algorithm with compact and parallel techniques, and applied it to the path planning. The results show that the proposed algorithm can achieve effective execution results *Zhang, Pu & Si (2021)* enhanced the ACO algorithm with an adaptive strategy for path planning, and the performance of path planning was significantly improved. *Quan et al. (2021)* discussed a method with self-adaptive harmony search (HS) algorithm and Morphin algorithm, and applied it to dynamic path design. *Zhang, He & Yang (2021)* used multiple strategies to improve the performance of sparrow search algorithm (SSA) and employed it for path planning, and the proposed algorithm achieved a short path and fast convergence. *Pan et al. (2022)* proposed a golden eagle optimizer with personal example and mirror reflection learning, and experimental results show that the algorithm has a good performance. These metaheuristics achieve good paths in some environments, but the robustness and adaptability of the algorithms need to be improved in complex environments.

In 2019, *Heidari et al. (2019)* proposed a new algorithm named Harris hawks optimization (HHO) algorithm, which was inspired by the hunting behavior of Harris hawks. The HHO algorithm has excellent search performance and has been used in many fields (*Fan, Chen & Xia, 2020*; *Krishna konijeti & Bharathi, 2022*; *Li et al., 2021*; *Turabieh et al., 2021*). However, the HHO algorithm, like most intelligent algorithms, easily misses the global optimum search and is trapped in a local optimum during the iterative process (*Abdel-Basset, Ding & El-Shahat, 2020*; *Akdag, Ates & Yeroglu, 2020*; *Chen et al., 2020a*).

To improve the path quality and avoid obtaining local optimal solutions, this work proposes a new method with a modified Harris hawks optimization (MHHO) algorithm for path planning. Firstly, to smooth the optimized path and reduce the path length, a linear path strategy (LPS) is used for path planning, which effectively straightens the corner segments of the path. Secondly, to avoid falling into the local optimum, a local search update strategy is used to obtain the global optimum. Finally, a nonlinear control strategy is employed for the HHO algorithm to improve the performance and convergence speed.

The rest of this article is as follows. The environment model, the problem of path planning, and the proposed algorithm are introduced in Section 2. Section 3 reveals the experiment simulations and discussion. Finally, Section 4 depicts the conclusions and future work.

## MATERIALS & METHODS

### Environment modeling and problem description

Typically, the grid method, topological map, and geometric method are used for environment modeling (*Chen et al., 2020b*; *Zhang et al., 2021*). In this article, the simulation environment is constructed using the grid method. The grid platform represents a two-dimensional environment where the mobile robot moves on the grid platform. On the platform, the moving area is represented by grid cells with binary information, indicated by "1" for the obstacle grid and "0" for the free grid. Figure 1 indicates a typical grid environment with a size of $20 \times 20$, where the blue grids denote obstacles and the white grids represent free space, respectively. When a mobile robot is in a white grid with no obstacles around, there are eight directions in which the mobile robot could move, as shown in Fig. 2.

For mobile robots, the grid-based path planning approach can be expressed as follows: beginning with the starting grid, path planning seeks to find a path that avoids the obstacle behavior to reach the end grid with a short path and in less time. In the process of optimization, the path is optimized by the path length. Consequently, the problem of path planning can be considered as an engineering optimization problem, and intelligent optimization algorithms could be used to address the problem.

### HHO algorithm

As a novel optimization algorithm, the HHO algorithm is also obtained by the evolution of nature, which mainly simulates the cooperative hunting process of Harris hawks. There are two phases in the HHO algorithm: the exploration phase and the exploitation phase (*Çetinbaş, Tamyürek & Demirtaş, 2021*; *Heidari et al., 2019*). The detailed process is expressed as follows.

The exploration phase is represented as *Heidari et al. (2019)* and *Kardani et al. (2021)*:

$$x(t+1) = \begin{cases} x_{\text{rand}}(t) - r_1 |x_{\text{rand}}(t) - 2r_2 x(t)|, & p \geq 0.5 \\ (x_{\text{prey}}(t) - x_m(t)) - r_3(L_b + r_4(U_b - L_b)), & p < 0.5, \end{cases} \tag{1}$$

$$x_{\text{avg}}(t) = \frac{1}{N} \sum_{i=1}^{N} x_i(t), \tag{2}$$

where $x(t)$ and $x(t+1)$ represent the position of the current iteration and the next iteration, respectively. $x_{\text{rand}}(t)$ denotes a randomly chosen position, $x_{\text{prey}}(t)$ represents the prey position, $L_b$ and $U_b$ represent the minimum and maximum values of the solution space, respectively. $r_1$, $r_2$, $r_3$, $r_4$, and $p$ represent the randomly assigned numbers in $(0,1)$, and $N$ is the number of total hawks.

The escape energy of each prey plays a critical role in the algorithmic search process, expressed as in Eq. (3). The value of the prey energy escaping determines whether the exploration phase ($|E| \geq 1$) or the exploitation phase ($|E| < 1$) is executed.

$$E = 2E_0(1 - t/T_{\max}), \tag{3}$$

where $E_0$ denotes the random initial escape energy, $t$ and $T_{\max}$ are the current and maximum number of iterations, respectively.

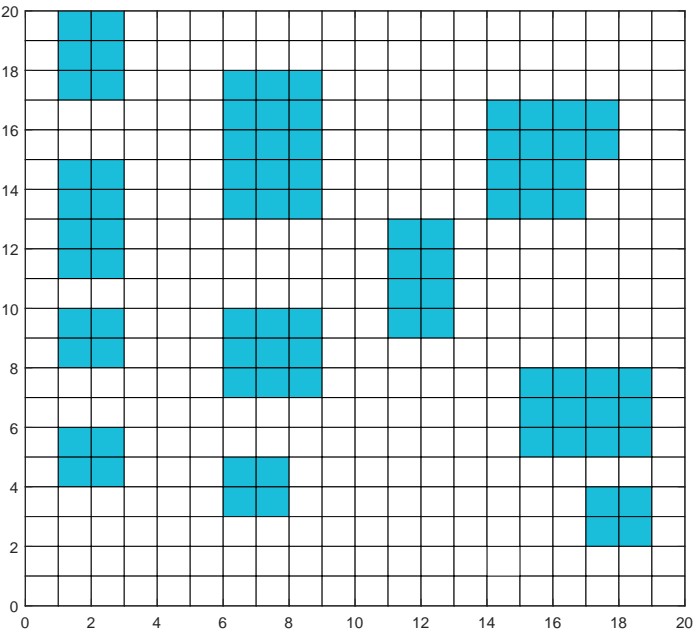

**Figure 1** **Grid map.**

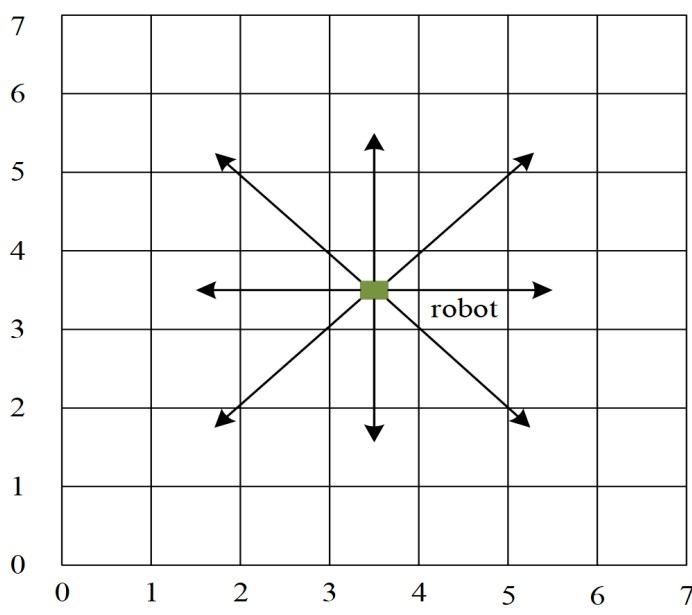

**Figure 2** **Possible directions of movement for a mobile robot.**

In the exploitation phase, the HHO algorithm implements four different processes, which are mainly the soft besiege, hard besiege, soft besiege with progressive rapid dives, and hard besiege with progressive rapid dives processes. Prey attempted to escape depending

on the environment, and $r$ is used to denote the probability of escape. The HHO algorithm employs a soft besiege to move closer to prey when the prey has enough escape energy ($|E| \geq 0.5$) but is still unable to escape from the encirclement ($r \geq 0.5$). The process is represented as:

$$x(t+1) = \Delta x(t) - E\left|J_{\text{prey}}x_{\text{prey}}(t) - x(t)\right|, \tag{4}$$

$$\Delta x(t) = x_{\text{prey}}(t) - x(t), \tag{5}$$

$$J_{\text{prey}} = 2(1 - r_5), \tag{6}$$

where $r_5$ denotes a random number, and $J_{\text{prey}}$ is the energy of jumping.

In the hard besiege process, the rabbit consumes so much energy that it does not have sufficient energy to escape ($r \geq 0.5$ and $|E| < 0.5$).

$$x(t+1) = x_{\text{prey}}(t) - E|\Delta x(t)|. \tag{7}$$

For the soft besiege with progressive rapid dives process, the rabbit retains enough strength to escape from the capture ($r < 0.5$ and $|E| \geq 0.5$). The process is expressed as:

$$Y_1 = x_{\text{prey}}(t) - E\left|J_{\text{prey}}x_{\text{prey}}(t) - x(t)\right|, \tag{8}$$

$$Z_1 = Y_1 + S \times \text{Levy}(n), \tag{9}$$

$$x(t+1) = \begin{cases} Y_1, & \text{fitness}(Y_1) < \text{fitness}(Z_1) \\ Z_1, & \text{fitness}(Y_1) > \text{fitness}(Z_1) \end{cases}, \tag{10}$$

where $S$ indicates a random-generation vector, Levy represents the Levy flight function (*Iacca, Dos Santos Junior & Veloso de Melo, 2021*), $n$ is the dimension of the problem, and *fitness* represents the fitness function.

For the hard besiege with progressive rapid dives process, the energy of prey is almost depleted and the prey is unable to escape safely. The process is expressed as:

$$x(t+1) = \begin{cases} Y_2, & \text{fitness}(Y_2) < \text{fitness}(Z_2) \\ Z_2, & \text{fitness}(Y_2) > \text{fitness}(Z_2) \end{cases}, \tag{11}$$

$$Y_2 = x_{\text{prey}}(t) - E\left|J_{\text{prey}}x_{\text{prey}}(t) - x_{\text{avg}}(t)\right|, \tag{12}$$

$$Z_2 = Y_2 + S \times \text{Levy}(n). \tag{13}$$

### Modified HHO algorithm

A modified HHO algorithm with three strategies for path planning is proposed. The LPS strategy is used to optimize the path, which produces the smooth and short paths. The nonlinear control strategy is applied to the HHO algorithm to enhance the convergence speed. The local search update strategy is used for the HHO algorithm to improve the convergence accuracy.

1. Linear path strategy

The LPS strategy means linearly achieving path planning as much as possible, which can generate a high-quality path and reduce the algorithm runtime (*Fareh et al., 2020*; *Zhang, He & Yang, 2021*). The process of LPS mainly completes the search for obstacles and the optimization of paths, as displayed in Fig. 3. The details are illustrated as follows.

Step 1: Start from the beginning of the path and generate three points sequentially.

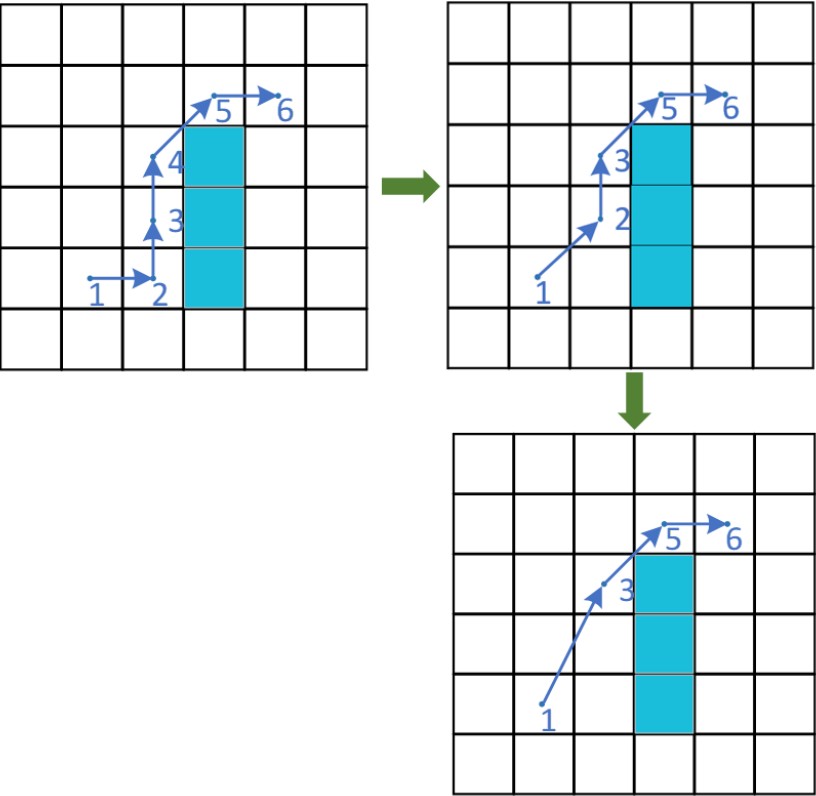

**Figure 3  The process of the LPS.**

Step 2: Calculate the distance of the generated points, determine the distance range, and search for obstacles within that range.

Step 3: When there is no obstacle in this scope, the second point is removed and the first and third points are connected to generate the path. Otherwise, the process terminates and continues the linearization of the path.

2. Nonlinear control strategy

To speed the convergence and obtain a low algorithm runtime, a nonlinear control strategy is proposed for the HHO algorithm for path planning (*Chen, Wang & Zhao, 2020*). The control parameter is given by

$$\omega = 2 \times e^{-(8t/T_{\max})^2}. \tag{14}$$

After applying the nonlinear control strategy, Eqs. (8) and (9) are updated as follows:

$$Y_1 = \omega x_{\text{prey}}(t) - E\left|J_{\text{prey}}x_{\text{prey}}(t) - x(t)\right|, \tag{15}$$

$$Z_1 = \omega Y + S \times \text{Levy}(D). \tag{16}$$

Moreover, Eqs. (12) and (13) are changed as follows:

$$Y_2 = \omega x_{\text{prey}}(t) - E\left|J_{\text{prey}}x_{\text{prey}}(t) - x_{\text{avg}}(t)\right|, \tag{17}$$

$$Z_2 = \omega Y_2 + S \times \text{Levy}(D). \tag{18}$$

3. Local search update strategy

For the robot path planning, the HHO algorithm easily obtains local optimal solutions and could not achieve the optimum of the path design. Thus, a local search update method is proposed to avoid being trapped in a local optimum. The local search update strategy focuses on randomly searching each dimension of the path and calculating the fitness value after the search. If the fitness value after the search is lower than the fitness value before the search, the previous path is updated with the search result. Otherwise, the search update path is discarded and the previous path is maintained.

The flowchart of the proposed algorithm with the three strategies is displayed in Fig. 4.

## Complexity analysis of MHHO algorithm

The computational complexity of HHO depends on three main processes: initialization, fitness calculation and individual position update. Assuming that population size is $N$, the maximum number of iterations is $T_{\max}$, and the search space dimension is $D$, the computational complexity of HHO can be calculated as follows (*Chen et al., 2020a*):

$$O(\text{HHO}) = O(N \times (1 + T_{\max} + T_{\max} \times D)). \tag{19}$$

Many aspects are mainly responsible for the computational complexity of the MHHO, which are initialization, fitness calculation, updating of individuals, LPS strategy, nonlinear control strategy and local search update strategy. The time complexity of LPS is $O(T_{\max} \times N^2)$. The time complexity of the local search update strategy is $O(T_{\max} \times N)$. The nonlinear control strategy only increases the calculation of the weights, and the increased computational complexity of the algorithm is negligible. Therefore, the whole computational complexity of MHHO is

$$O(\text{MHHO}) = O(N \times (1 + T_{\max} \times (2 + D + N))). \tag{20}$$

## Implement path planning for the MHHO algorithm

For mobile robots, the path planning for executing the MHHO algorithm is described as follows:

Step 1: The simulation environment model is constructed and optimization algorithm parameters are set. The path length is chosen as the fitness function for the optimization algorithm.

Assuming that the coordinates of a node of the path are denoted as $P_i(x_i, y_i)$, and the path length is given by

$$f_{\text{path\_length}} = \sum_{i=0}^{k} \sqrt{(x_{i+1} - x_i)^2 + (y_{i+1} - y_i)^2}, \tag{21}$$

where $k$ is a node of the path.

The calculation of path length is employed as the fitness function, and the LPS is also used to compare the lengths of the different paths during the optimization iterations to select the optimal path.

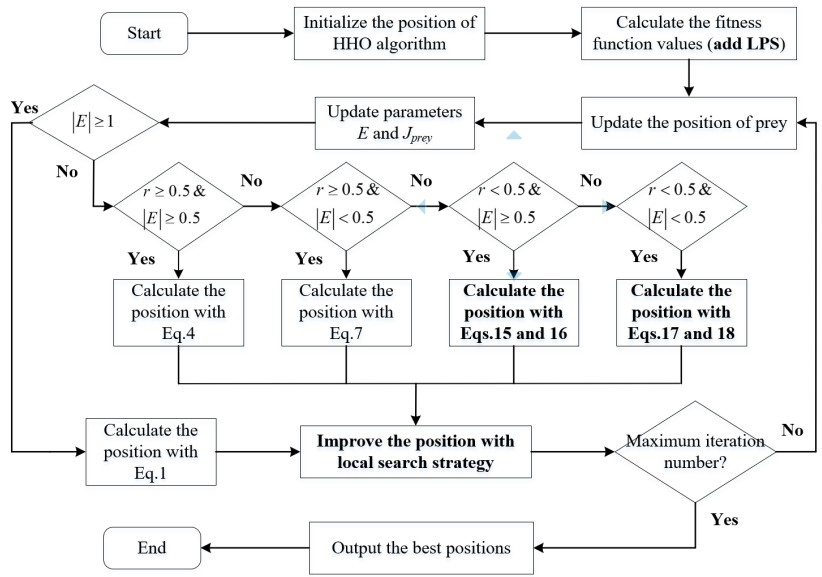

**Figure 4    Flowchart of the proposed algorithm.**

Step 2: Initialization of the proposed algorithm. The population individuals are randomly generated, the fitness function values of the population individuals are calculated, and the optimal individual is selected.

Step 3: Implementation of the exploration phase and the exploitation phase of the MHHO algorithm.

Step 4: The criteria for algorithm termination iterations. If the maximum number of iterations is reached, the algorithm stops iterating and outputs the optimal position. Otherwise, the algorithm continues to find the best position.

## EXPERIMENTAL ANALYSIS AND DISCUSSION

### Simulation environment and parameters setting

To validate the effectiveness of the proposed algorithm, simulation experiments are performed in two different environments. To effectively display the path finding results, the grid map size of the simulation environments is set to 20 × 20, as displayed in Fig. 1. In the grid map, the starting grid and the ending grid were at the left bottom corner and top right corners, respectively.

In this study, several optimization algorithms are applied to the path planning of robots. ACO, HHO, ISSA, and MHHO were simulated under the same conditions. All simulations were carried out using a Huawei computer equipped with AMD Ryzen 5 @ 3 GHz and 16 GB of RAM. All algorithms were simulated employing MATLAB R2018b software. To ensure the effective comparison of the algorithms, all algorithms have the same population size and iteration numbers. The parameters of each algorithm are listed in Table 1.

**Table 1  Main parameters of each algorithm.**

| Algorithm | Parameter | Value |
|---|---|---|
| ACO | Population size | 20 |
| | Maximum iteration number | 100 |
| | Pheromone factor | 2 |
| | Heuristic factor | 6 |
| | Pheromone evaporation factor | 0.1 |
| ISSA | Population size | 20 |
| | Maximum iteration number | 100 |
| | Proportion of discoverer | 0.3 |
| | Proportion of scout | 0.2 |
| | Safety value | 0.8 |
| HHO | Population size | 20 |
| | Maximum iteration number | 100 |
| MHHO | Population size | 20 |
| | Maximum iteration number | 100 |

## Evaluation criterion

Path planning for mobile robots aims at the best time and shortest path, so it is necessary to use several indicators for path evaluation. We chose two indicators to evaluate the path quality: the optimum path and the execution time (*Deng et al., 2021*; *Montiel, Sepúlveda & Orozco-Rosas, 2014*).

The optimum path is the length of the best search path obtained by the algorithm. The execution time of the algorithm is the time taken by the algorithm to achieve the optimal path.

## Simulation results analysis

To prove the stability and efficiency of the proposed algorithm, experiments with the MHHO, HHO, ACO, and ISSA were carried out in the same environment. The results of ACO, HHO, ISSA, and MHHO algorithms in Environment 1 are listed in Table 2. Figures 5 and 6 respectively show the optimum path diagram and the convergence curve with different algorithms in Environment 1. From Table 2, the execution time of ACO, HHO, ISSA, and MHHO algorithms are 1.97 s, 0.41 s, 1.23 s, and 0.89 s, respectively. From Fig. 5, the path achieved with the MHHO algorithm has good path length and path smoothness compared with ACO, HHO, and ISSA algorithms. As displayed in Fig. 6, the MHHO algorithm achieves high convergence accuracy when the number of iterations is small. The MHHO algorithm has the fastest convergence speed compared to the ACO, HHO, and ISSA algorithms.

In Environment 2, the experiment was performed with the same parameters, and the results of different algorithms are listed in Table 3. The optimum path chart and the convergence curve are shown in Figs. 7 and 8, respectively. From Table 3, the execution time of the ACO, HHO, ISSA, and MHHO algorithms are 1.45 s, 0.38 s, 0.76 s, and 0.69 s,

**Table 2  Results of different algorithms in Environment 1.**

| Algorithm | Optimum path | Execution time |
| --- | --- | --- |
| ACO | 30.63 | 1.97 s |
| HHO | 29.41 | 0.41 s |
| ISSA | 28.73 | 1.23 s |
| MHHO | 27.90 | 0.89 s |

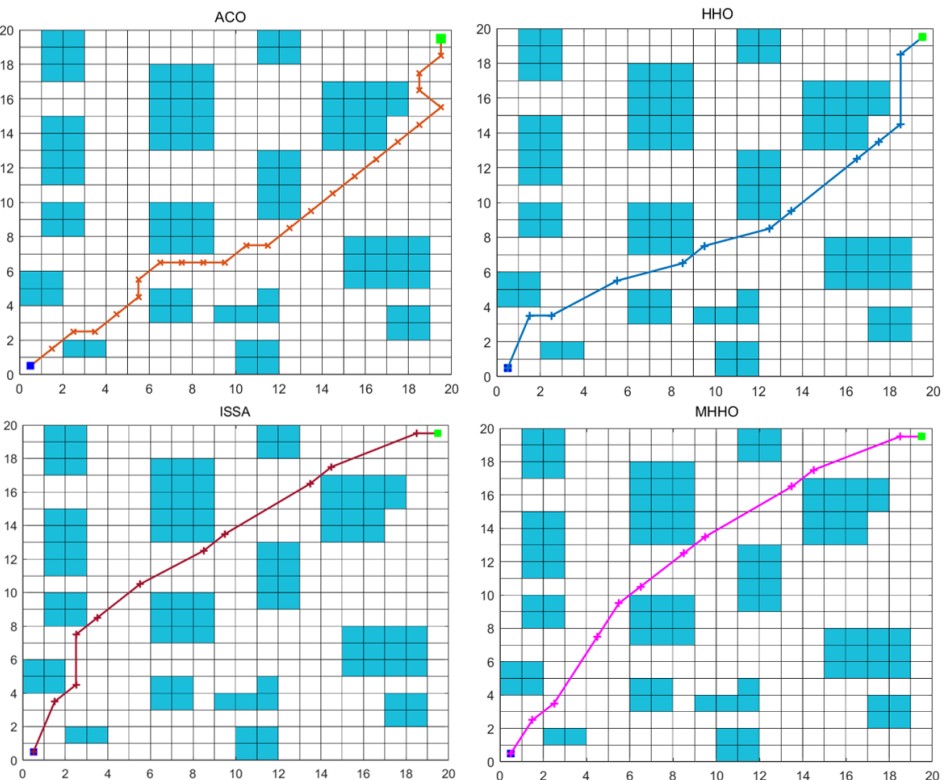

**Figure 5  The optimum path generated by different algorithms in Environment 1.**

**Table 3  Results of different algorithms in Environment 2.**

| Algorithm | Optimum path | Execution time |
| --- | --- | --- |
| ACO | 34.63 | 1.45 s |
| HHO | 32.33 | 0.38 s |
| ISSA | 29.74 | 0.76 s |
| MHHO | 29.41 | 0.69 s |

respectively. From Figs. 7 and 8, the optimum path and the convergence rate of the MMHO algorithm are also superior to the ACO, HHO, and ISSA algorithms.

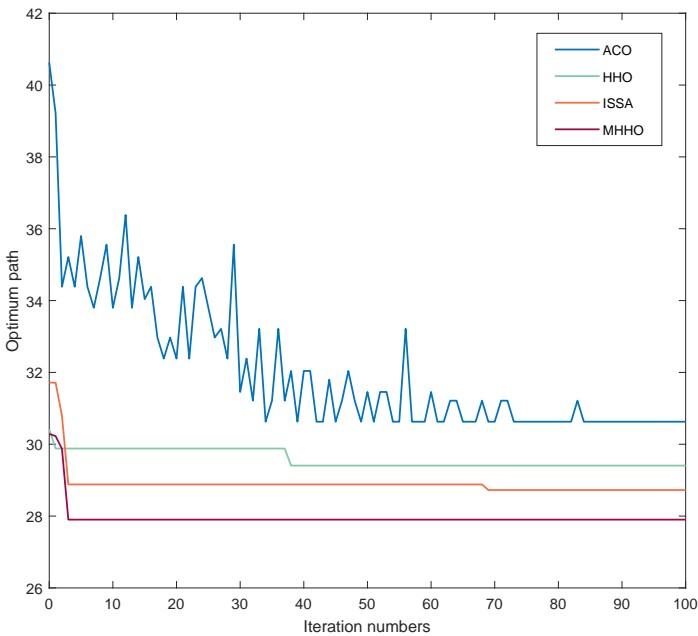

**Figure 6** Convergence curve of different algorithms in Environment 1.

## Performance comparison

Swarm intelligence algorithm has a certain randomness when used to address practical engineering problems. To prove the adaptability and robustness of the proposed algorithm, the simulation experiments of the ACO, HHO, ISSA, and MHHO algorithms are conducted 50 times independently in Environments 1 and 2, respectively. The indicators of the optimum path and the implementation time of the four algorithms are achieved.

Figures 9 and 10 display the results of the four algorithms with the two indicators in Environment 1. As seen in Fig. 9, the MHHO algorithm obtains the optimal path and the length of the path has small fluctuations. The paths obtained by the HHO algorithm have large fluctuations. The optimum path of the MHHO algorithm outperforms other algorithms. From Fig. 10, the ACO algorithm has longest execution time and the HHO algorithm has the shortest execution time compare with other algorithms. The implementation time of the MHHO algorithm is slightly increased concerning the HHO algorithm due to the inclusion of multiple strategies in the algorithm.

In Environment 2, the results of the four algorithms are displayed in Figs. 11 and 12. From Fig. 11, the length of obtaining the optimal path is increased due to the increase in the number of obstacles. The MMHO algorithm still has a good optimum path compare with other algorithms. As displayed in Fig. 12, the MMHO and ISSA algorithms have similar execution times, and the HHO algorithm achieves the minimum execution time.

To further discuss the capabilities of the MHHO algorithm, we count the test results of the path planning in different environments. Table 4 shows the results of the algorithms in different environments. In Environment 1, the MHHO algorithm has the best performance for the optimum path, and ISSA algorithms have similar execution times, and the HHO

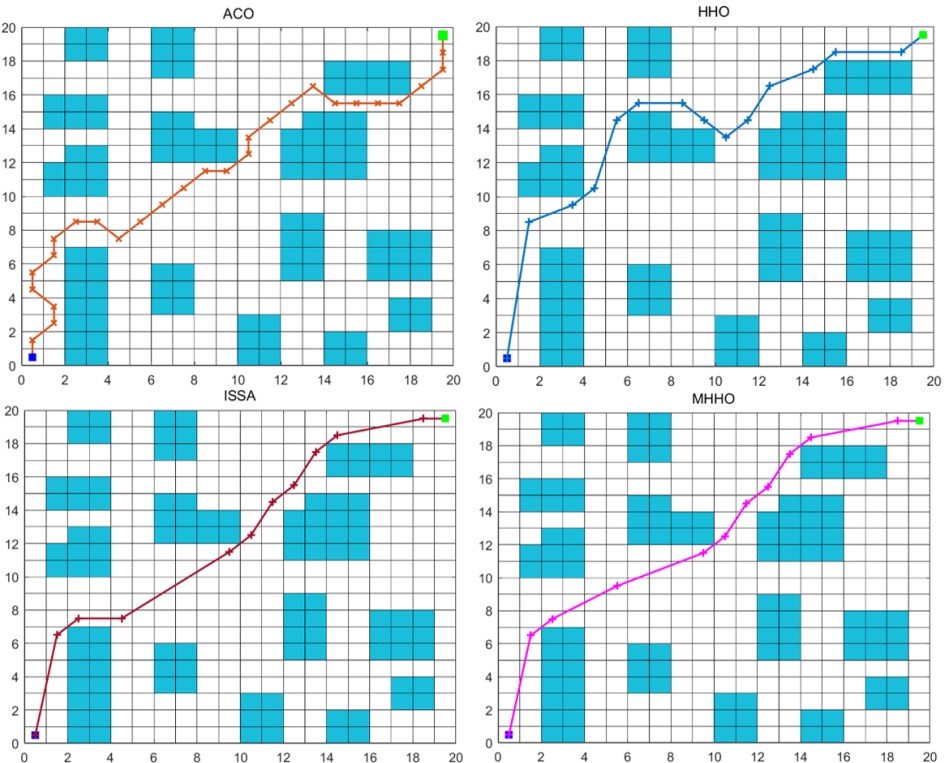

**Figure 7  The optimum path generated by different algorithms in Environment 2.**

algorithm has the minimum execution time. In Environment 2, the MHHO algorithm also obtains the optimal path. In addition, the MHHO algorithm outperforms the ISSA algorithm in terms of path optimization and algorithm execution time. Therefore, the MHHO algorithm is an effective method for robotic path design.

## CONCLUSIONS

For path planning of mobile robots, traditional algorithms are easily trapped in localization and produce unsmooth paths. In this study, an MMHO algorithm with LPS, non-linear control strategy, and local search update strategy is presented to solve the above challenges. The proposed algorithm is characterized by fast convergence and strong optimization capability. To validate the performance of the algorithm, the experiments were carried out under different environments, and the optimal path and the execution time are selected to verify the path quality. Compared with the ACO and ISSA algorithms, the MHHO algorithm has the shortest path length and the least execution time. Although the implementation time of the MHHO algorithm is more than that of the HHO algorithm, the optimum path length is significantly better than that of the HHO algorithm. Therefore, the proposed algorithm is an effective method for robot path planning. In future work, we will further optimize the MHHO algorithm and demonstrate the performance in a real dynamic environment.

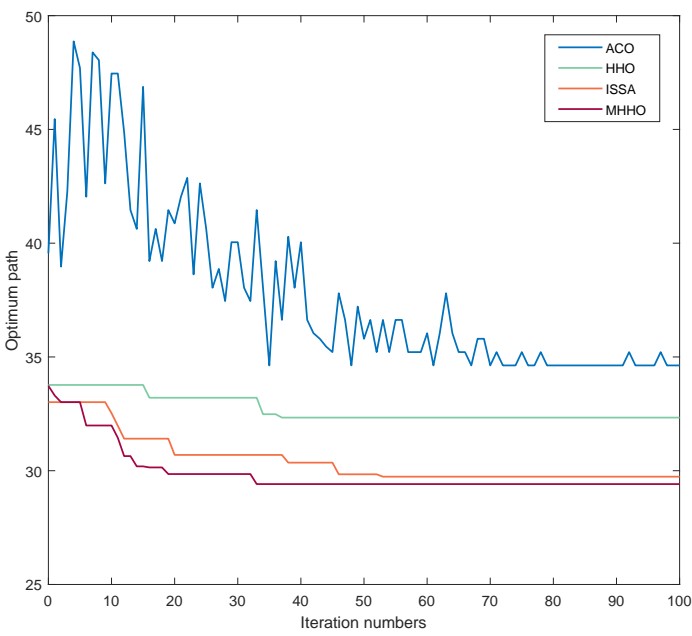

**Figure 8** Convergence curve of the different algorithms in Environment 2.

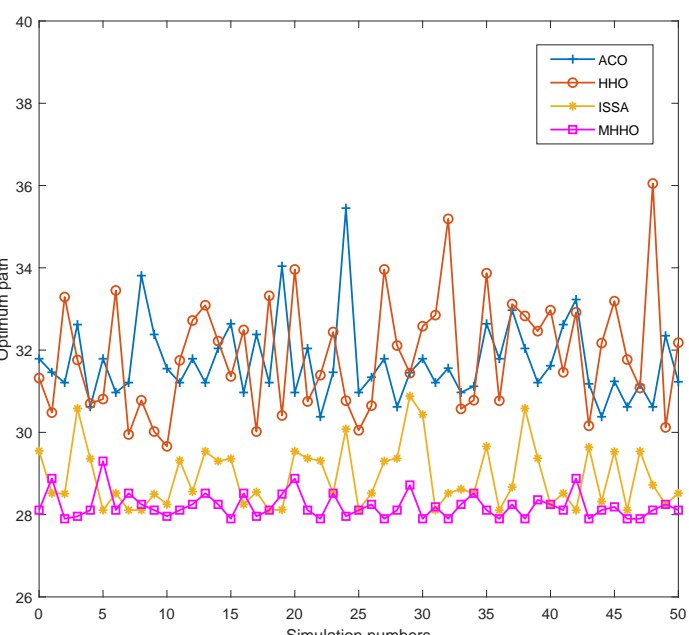

**Figure 9** The optimum path of different algorithms in Environment 1.

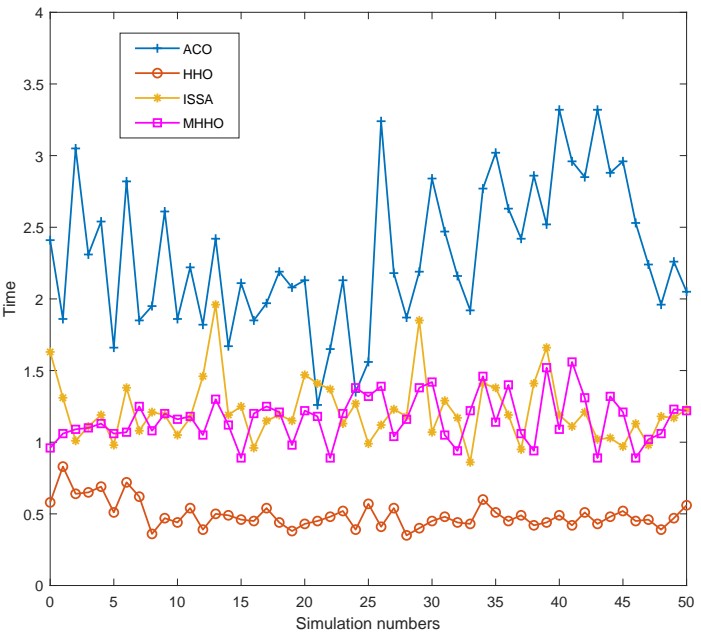

**Figure 10** The execution time of different algorithms in Environment 1.

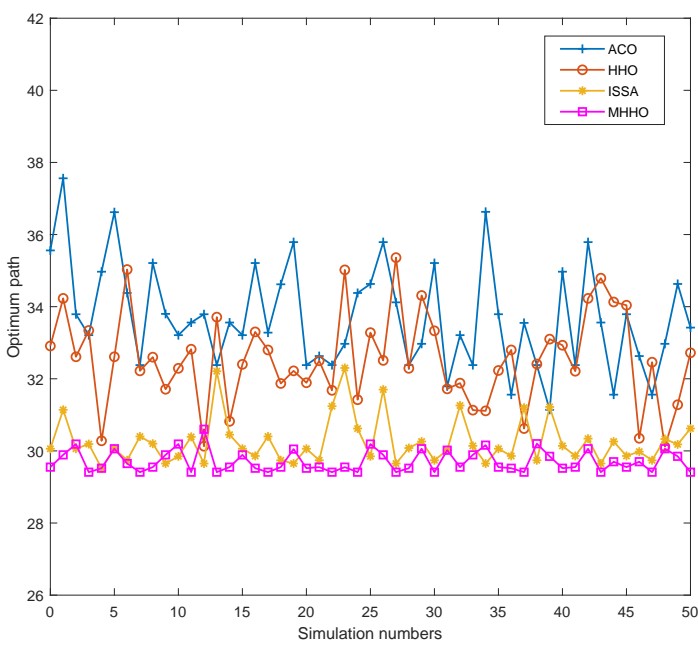

**Figure 11** The optimum path of different algorithms in Environment 2.

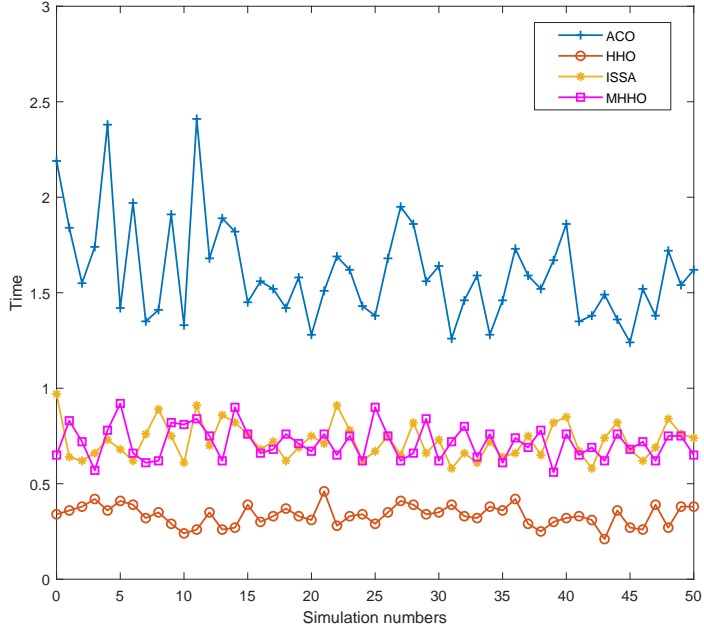

**Figure 12  The execution time of different algorithms in Environment 2.**

**Table 4  The results of different algorithms in Environments 1 and 2.**

| Environment | Evaluation criterion | | ACO | HHO | ISSA | MHHO |
|---|---|---|---|---|---|---|
| Environment 1 | Optimum path | Maximum | 35.45 | 36.05 | 30.88 | 29.30 |
| | | Minimum | 30.38 | 29.66 | 28.11 | 27.90 |
| | | Average | 31.71 | 31.93 | 28.90 | 28.20 |
| | Algorithm execution time (s) | Maximum | 3.32 | 0.83 | 1.96 | 1.56 |
| | | Minimum | 1.26 | 0.35 | 0.86 | 0.89 |
| | | Average | 2.31 | 0.48 | 1.21 | 1.17 |
| Environment 2 | Optimum path | Maximum | 37.56 | 35.36 | 32.30 | 30.60 |
| | | Minimum | 31.14 | 30.12 | 29.52 | 29.41 |
| | | Average | 33.60 | 32.50 | 30.23 | 29.71 |
| | Algorithm execution time (s) | Maximum | 2.41 | 0.46 | 0.97 | 0.92 |
| | | Minimum | 1.24 | 0.21 | 0.58 | 0.56 |
| | | Average | 1.59 | 0.33 | 0.74 | 0.71 |

## ACKNOWLEDGEMENTS

We sincerely appreciate the authors of the references for their research work. We are also grateful for the help provided by the Intelligent Networked Vehicle Key Technology Laboratory of Lu'an City.

### Funding

This research was funded by the Provincial Natural Science Research Project of Anhui University (Grant Nos. 2022AH051667 and 2022AH051669), the Natural Science Key Scientific Research Project of West Anhui University (Grant Nos. WXZR202103 and WXZR202004). The funders had no role in study design, data collection and analysis, decision to publish, or preparation of the manuscript.

### Grant Disclosures

The following grant information was disclosed by the authors:
Provincial Natural Science Research Project of Anhui University: 2022AH051667, 2022AH051669.
Natural Science Key Scientific Research Project of West Anhui University: WXZR202103, WXZR202004.

### Competing Interests

The authors declare there are no competing interests.

### Author Contributions

- Cuicui Cai conceived and designed the experiments, performed the experiments, analyzed the data, performed the computation work, authored or reviewed drafts of the article, and approved the final draft.
- Chaochuan Jia conceived and designed the experiments, performed the experiments, performed the computation work, prepared figures and/or tables, and approved the final draft.
- Yao Nie analyzed the data, performed the computation work, authored or reviewed drafts of the article, and approved the final draft.
- Jinhong Zhang conceived and designed the experiments, authored or reviewed drafts of the article, and approved the final draft.
- Ling Li analyzed the data, prepared figures and/or tables, and approved the final draft.

### Data Availability

The code is available at Github and Zenodo:

https://github.com/Caihappy/MMHO-algorithm.git.

Caihappy. 2023. Caihappy/MMHO-algorithm: Version 1.0 (v1.0). Zenodo. https://doi.org/10.5281/zenodo.8022612.

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
