# Peer review of "A path planning method using modified harris hawks optimization algorithm for mobile robots"

_PeerJ Computer Science, doi:10.7717/peerj-cs.1473_

## Round 0.1 · original submission · Major Revisions

Based on reviewer's comments, the manuscript needs a major revision.

Reviewer 1 ·

Basic reporting

NIL

Experimental design

NIL

Validity of the findings

NIL

Additional comments

The overall chapter organization of the paper is relatively good. However, these are still the following problems that need to be resolved:
1.In the abstract, the modified harris hawks optimization(MHHO) algorithm you proposed is better than ant colony optimization (ACO) algorithm, improved sparrow search (ISSA) algorithm, and harris hawks optimization(HHO)in terms of convergence speed. How did you come to this conclusion? Have you used the CEC test function to test the excellent performance of your improved algorithm in terms of the convergence speed and the convergence accuracy?

2.In the introduction of your article, you have stated that HHO is prone to falling into local optima. Can you explain why HHO is prone to falling into local optima? Is it because the structure of the algorithm is set unreasonably? Is there any relevant article that can explain this issue?

3.In your MHHO, the nonlinear control strategy you proposed is similar to the inertia weight strategy in particle swarm optimization (PSO) algorithm. Why don't you apply PSO to robot path planning?

4.In the environment modeling section, the moving area is represented by grid cells with binary information, indicated by "1" for the obstacle grid and "0" for the free grid. Similar coding methods can also be used in genetic algorithm (GA). Why don't you use GA to solve this problem?

5. There are many algorithms related to path planning method such as
5.1 Golden eagle optimizer with double learning strategies for 3D path planning of UAV in power inspection
5.2 A Parallel Compact Cuckoo Search Algorithm for Three-Dimensional Path Planning
The authors may introduce those existing methods so as to broaden the scope of this paper.

Reviewer 2 ·

Basic reporting

This work proposes a path planning method using a modified Harris hawks optimization (MHHO) algorithm to address the problem of path planning. The manuscript is well written however, it needs further improvement before it can be accepted for review.
1. The experimental setup along-with a description of simulation software need to be clearly mentioned.
2. The authors must elaborate why they have used 20 x 20 grid to for the simulation? why not more? why not less?
3. The algorithm complexity analysis (time and space) especially in the worst case scenario needs to be conducted and represented.
4. A section "comparison" must be added in the manuscript representing and reasoning about the results of comparison among other similar algorithms.
5. The authors claim that the proposed algorithm tackles the local maxima problem, however they were unable to address this issue in the manuscript.
6. The figures' captions need to be carefully reviewed.
7. The reasoning behind the achieved experimental results must be explained.
8. Perhaps a graph/figure showing only the modification the authors have proposed in the original algorithm may be more helpful for the readers.
9. I would like to see the experimental results in case where the start and end are chosen randomly.

Experimental design

The experimental setup along-with a description of simulation software need to be clearly mentioned.

Validity of the findings

4. A section "comparison" must be added in the manuscript representing and reasoning about the results of comparison among other similar algorithms.
5. The authors claim that the proposed algorithm tackles the local maxima problem, however they were unable to address this issue in the manuscript.

---

## Round 0.2 · accepted · Accept

The authors addressed the concerns that reviewers had. The manuscript is in the publishable shape.